# Translating Co-Design from Face-to-Face to Online: An Australian Primary Producer Project Conducted during COVID-19

**DOI:** 10.3390/ijerph18084147

**Published:** 2021-04-14

**Authors:** Alison Kennedy, Catherine Cosgrave, Joanna Macdonald, Kate Gunn, Timo Dietrich, Susan Brumby

**Affiliations:** 1National Centre for Farmer Health, Deakin University, Hamilton, VIC 3300, Australia; susan.brumby@wdhs.net; 2Adjunct Research Fellow, New England Institute of Healthcare Research, School of Rural Medicine, University of New England, Armidale, NSW 2350, Australia; ccosgra2@une.edu.au; 3National Centre for Farmer Health, Western District Health Service, Hamilton, VIC 3300, Australia; joanna.macdonald@wdhs.net; 4Department of Rural Health, Allied Health and Human Performance, University of South Australia, Adelaide, SA 5000, Australia; kate.gunn@unisa.edu.au; 5Griffith Business School, Griffith University, Brisbane, QLD 4001, Australia; t.dietrich@griffith.edu.au

**Keywords:** online co-design, mental health, primary producer, farmer, fisher, COVID-19, risk prevention

## Abstract

Primary producers face considerable risks for poor mental health. While this population can be difficult to engage in programs to prevent poor mental health, approaches tailored to reflect the context of primary producers’ life and work have been successful. This paper reports on the co-design phase of a project designed to prevent poor mental health for primary producers—specifically, the advantages, challenges and considerations of translating face-to-face co-design methods to an online environment in response to COVID-19 restrictions. The co-design phase drew upon the existing seven-step co-design framework developed by Trischler and colleagues. Online methods were adopted for all steps of the process. This paper models how this co-design approach can work in an online, primary producer context and details key considerations for future initiatives of this type. The development of online co-design methods is an important additional research method for use not only during a pandemic but also when operating with limited resources or geographic constraints. Results demonstrate the following: (i) co-designing online is possible given adequate preparation, training and resource allocation; (ii) “hard to reach” populations can be engaged using online methods providing there is adequate early-stage relationship building; (iii) co-design quality need not be compromised and may be improved when translating to online; and (iv) saved costs and resources associated with online methods can be realigned towards intervention/service creation, promotion and user engagement. Suggestions for extending Trischler and colleagues’ model are incorporated.

## 1. Introduction

Primary producers (those involved in the business of farming, fishing or forestry) face a large number of work-related risk factors that contribute to poor mental health. These include isolation (social and geographic), market uncertainty, commodity pricing, climatic extremes and regulatory constraints. This exposure to many factors beyond their control leads to uncertainty and the need to address knowledge and skill deficits and strengthen wellbeing and the individual’s sense of job control [1,2,3,4]. While programs preventing poor mental health are deemed important, this has traditionally been difficult to achieve with primary producers—a population described as “a lost tribe” when it comes to engagement around health and wellbeing [5]. Responses to poor mental health have often lacked careful consideration of both expert and user input and have been characterized by limited wider stakeholder consultation [6], thus limiting capacity to build solutions that generate support and encourage engagement. Involving primary producers, experts and other relevant stakeholders through co-design may provide further opportunities to increase engagement with the co-design of mental health prevention programs and increase the uptake and general acceptance of these types of programs.

The COVID-19 pandemic has amplified citizens’ use of the Internet, with usage increasing by between 40% and 100% compared to pre-lockdown levels [7]. Researchers have needed to pivot online to continue their citizen-focused research due to the limitations caused by the pandemic. All co-creation methods, including co-design, have historically strongly relied on face-to-face engagement, and translating these methods into an online format is now urgently warranted. This point in time presents a valuable opportunity for exploring new ways of empowering primary producers to work with researchers to identify the challenges and co-design the best solutions [8]. This also provides the opportunity to fill an important gap for co-design research and demonstrate how co-design can be executed fully online using a sequential and evolving series of co-design workshops following the process of Trischler et al. This paper makes important theoretical and methodological contributions, guiding future researchers and practitioners to run sequential co-design sessions online with multiple and cross-sectoral stakeholder groups.

This paper reports on the process of adapting an existing seven-step co-design framework [9]—designed for face-to-face engagement—to an online delivery mode for The Primary Producer Knowledge Network (PPKN) project undertaken in the state of Victoria, Australia. Victoria is the smallest and most populous state on Australia’s mainland. Agricultural land occupies around 56% of the state and produces 26% of the total gross value of agricultural production in Australia [10]. The adaption resulted from the inability to meet face-to-face due to the COVID-19 pandemic and subsequent lockdown of the state of Victoria in response to the same [11]. This paper is guided by the following research questions: (a) How could online co-design, informed by the co-design framework of Trischler and colleagues [9] be employed in a Victorian primary producer context? (b) What are the advantages, challenges and considerations of using this approach?

## 2. Materials and Methods

### 2.1. Context

The PPKN is led by the National Centre for Farmer Health (NCFH). The NCFH encompasses research, service delivery and education and aims to improve the health, wellbeing and safety of primary producers, their workforce, their families and rural communities across Australia and globally. The aim of the PPKN is to develop strategies to prevent workplace-related factors adversely affecting the mental health of primary producers, their workforce and their families, in collaboration with primary producers and industry stakeholders, using co-design methods. The co-design phase of the PPKN will inform the development and delivery of a range of potential face-to-face and digital strategies and resources such as a website, interactive capacity building program and social media. The project timeline for the PPKN is outlined in Figure 1.

Due to increasing case numbers of COVID-19, Victoria declared a “state of emergency” under the Public Health and Wellbeing Act 2008 (Vic) imposing restrictions on individual freedoms and movement. Responding to these significant restrictions and lockdowns, additional guidelines were established by many Human Research Ethics Committees (in Australia, over 200 Human Research Ethics Committees review all research proposals involving human participants and ensure they meet the guidelines outlined in the National Statement on Ethical Conduct in Human Research, 2007) regarding the conduct of research. These changes commonly removed all face-to-face interactions for research participants and staff collecting data and stopped all travel. This study’s research team viewed this as an opportunity to pivot the guiding co-design framework (Trischler et al., 2019) from a face-to-face to an online approach. The co-design phase of the project was approved by Deakin University’s Faculty of Health Ethics Committee (HEAG-H 79_2020).

Acknowledging the reflexive relationship between researchers and the research process, consideration of the researchers’ role is considered imperative. The PPKN research team is interdisciplinary and cross-sectoral, comprising six academics (the authors of this paper). The first phase of the project is being guided by a leading international authority (T.D.) on co-design methodology and its application for building engaging strategies to produce effective behavior change campaigns with a positive impact on target communities. The research team has broad research, clinical and life experience. Five of the six researchers (A.K., C.C., J.M., K.G., S.B.) are living rurally across four states and/or have rural backgrounds. This group has extensive applied research experience in rural health and wellbeing. Three members (A.K., K.G., S.B.) have specialist expertise in primary producer health and wellbeing in addition to agricultural expertise. All team members have experience working collaboratively with rural communities using community-based participatory research approaches including co-design.

### 2.2. Pre-Workshop Phase

The co-design phase of the study drew upon the existing seven-step co-design framework (see Figure 2) developed by Trischler and colleagues [9] (including T.D.). The framework has been proven appropriate for engaging with participants who may be reluctant or difficult to engage [12]—such as primary producers on the topic of mental health.

The seven steps guiding this co-design process are as follows:(1)Resourcing (sourcing relevant information and gaining understanding of the problem to be addressed);(2)Planning (working with industry groups and other stakeholders to coordinate recruitment and workshop facilitation, planning for unforeseen challenges, e.g., COVID-19);(3)Recruiting (direct invitation of innovative and engaged primary producers and stakeholders and widening the recruitment net through a public expression of interest process);(4)Sensitizing (familiarizing participants with the context and encouraging reflection);(5)Facilitation (activities and/or discussions followed by creative development of ideas);(6)Reflecting (testing ideas for originality, feasibility, user value, transformative value of workplace systems and likelihood of preventing risks to wellbeing);(7)Building for Change (collaborative and iterative effort to build viable solutions that receive user and stakeholder support).

## 3. Results

Results, in the form of adapted methods and lessons from the three discrete, yet interconnected, online co-design phases (pre-workshop, workshop and post-workshop) are detailed below. This includes an overview of each co-design step within the phases, details on how this method can be employed in an online environment with primary producers and considerations arising from this transition. In addition, the shift to an online environment meant that broad adaptions to project management procedures had to be made. This section begins with a brief discussion of the general project management approach.

The research team met on a fortnightly basis via Zoom [13] to discuss online co-design planning. Zoom was the chosen communications platform due to its demonstrated reliability in providing quality video and online conferencing functionality. At the time of project planning, Zoom offered all of the functions required by the research team, including breakout rooms, poll features, recording capability and the ability to share screens. The lead research institution also held a licensed account, enabling access to security functions such as password-protected meeting spaces. Meeting planning included developing running sheets to maintain focus and establish realistic objectives. Prior to the adoption of online co-design methods, project planning had envisaged drawing on the skills of only three of the team members (A.K., J.M., T.D.) to facilitate the co-design process. This was due to the spread of research team members across four states and the limited availability of project resources to fund extensive travel. Using digital methods of engagement allowed all members of the research team to easily contribute to all aspects of the co-design process, enabling a richer understanding of the research problem, the co-design process and the strategies to ensure successful engagement and execution. Team members leveraged off each other’s skills and capacities to ensure that project milestones were met without any delays to the original planning process stage. The research team worked effectively together online to overcome any time, distance and communication problems.

Quarterly Advisory Group meetings were also held using the Zoom platform. For those Advisory Group members directly involved in primary production, using online engagement methods reduced the time taken away from production tasks and provided them with the opportunity to continue critical tasks while still participating in meetings (e.g., one Advisory Group member remained working in his tractor while linked into the meeting). A paid Zoom subscription ensured access to secure, password-protected meetings. The online format eliminated the time and cost required for travel that is usually associated with contributing to rurally based projects and reduced transcription costs due to Zoom’s auto-transcribe feature. Budgets were re-allocated to enhance other project activities (e.g., after co-design content development).

### 3.1. Pre-Workshop Phase

The pre-workshop phase included an iterative process of resourcing, planning and recruiting.

#### 3.1.1. Resourcing, Planning and Recruitment

An understanding of the problem to be addressed was achieved via a rapid review of the literature. This was enabled by the expertise of several team members (A.K., K.G., S.B.) with strong understanding of the existing literature already available. The review of the literature was contextualized by phone interviews conducted with 10 industry stakeholders (representing a broad range of primary production sectors such as horticulture, grazing, fishing, dairy and cropping) and with in-depth knowledge of the challenges faced by primary producers. The interview schedule was informed by the literature review and explored six overarching categories of workplace-related mental health risk factors (business inputs; work arrangements; occupational hazards; uncertainty about business viability; government and public opinion; and physical, mental and interpersonal factors). Stakeholders were asked to rate the specific factors identified in each category relative to their contribution to the poor mental health of primary producers and were provided an opportunity to explain these ratings. Ratings were made on a 5-point Likert scale [14] ranging from 0 (not at all important) to 4 (very important). A “not applicable” option was also included to allow for the varied mental health risk factors present across different primary production sectors (e.g., fishing versus dairy farming). Stakeholders were also asked to describe any programs they were aware of that had been developed to reduce risks to mental health within their primary production sector. The purpose of this was to identify programs that may not have been captured in the literature but may warrant consideration in the current project. From a combined understanding of the literature, the interview data and pre-existing researcher expertise, the research team developed a series of 10 “activity cards” (see Table 1). These were representations of potential activities/programs designed to reduced risks to mental health (see Section 3.2.1 for how these cards were used in the co-design process). Each card comprised a title, brief description and representative image.

A purposive sampling strategy [15] was adopted for recruitment of the co-design participants, ensuring the inclusion of information-rich participants and targeting the specific area of interest. This was achieved using two methods:A call for expressions of interest made via the National Centre for Farmer Health ENews and social media platforms, the Farmer Health website www.farmerhealth.org.au (accessed on 12 July 2020), rural media outlets and digital communications channels of existing health and primary industry networks.Inviting Advisory Group members to nominate key individuals in their industry sector with knowledge and experience of specific workplace factors contributing to the risk of poor mental health.

The participants were purposively chosen to ensure a spread of industry type, gender, geographic location and age. The first co-design workshop comprised primary producers (*n* = 12), and the second comprised service providers and industry stakeholders (*n* = 11).

#### 3.1.2. Online Considerations for the Pre-Workshop Phase

Researchers were aware that a lack of previous experience, increased age, poor resources and limited training have been reported as influencing the understanding and use of information and communication technology [16]. These factors were therefore likely to affect the adaptation to an online co-design environment. However, it was also viewed that the entirely online format potentially reduced the barriers to participation that rural participants frequently experience when involved in city- or interstate-based projects. As a single online participant contributing to a largely face-to-face meeting audience, contributions can feel tokenistic, unheard and discounted [17]. In this project, all participants (and researchers) were online, which we believe provided a more equal footing and significantly enhanced the quality of discussions. The online format also broadened the potential co-design participant group from those who have the capacity/funds/time/willingness to travel to a central face-to-face meeting point to those who had a genuine desire and purpose to participate. The online format also enabled equitable representation regardless of geographic location.

The COVID-19 social distancing regulations and subsequent dramatic shift of a range of daily tasks to an online digital environment (e.g., telehealth, education and training, working from home) meant that people had more awareness of online engagement, as well as greater willingness and expertise to engage online, than they did prior to the pandemic [7]. Victoria, in particular, witnessed a seismic shift in education delivery with all students, from prep to postgraduate, learning in an online environment for the major part of the 2020 education year [11]. There were many and various concerns expressed by students, parents, educators and the community that online learning would not provide the same benefits of peer support as traditional face-to-face teaching. According to Learning Designer David Seignior, online learning can be more interactive than face-to-face learning [18]. He describes that well-constructed online education actually privileges the social cohort experience by putting it at the forefront. Successful online learning requires sharing of knowledge, having a cognitive presence and having a teacher or facilitator curating to encourage engagement and collaboration. These online learning fundamentals are also required for successful online co-design, as outlined below.

### 3.2. Workshop Phase

For each of the workshops, the process of sensitizing and facilitating was repeated. Via the process of reflection following workshops 1 and 2, adaptations were made to the co-design process for the subsequent workshop, thus creating an iterative loop. This similarly influenced the process used in the feedback session (see Section 3.3.1).

#### 3.2.1. Sensitizing

Sensitizing activities were disseminated a week in advance of workshops 1 and 2 in the form of an online Qualtrics [19] survey (communicated using a link within an explanatory email). The survey for workshop 1 (involving primary producers) and workshop 2 (involving stakeholders) commenced with the informed consent process. Consent involved agreeing to a series of statements about the nature of project contributions, confidentiality and data reporting/storage. An embedded link to the participant information form was also included.

Following their provision of informed consent, participants in workshop 1 were presented with the series of 10 activity cards including representative pictures to stimulate reflection and a brief description of activities that could potentially be included in a program of work to reduce risks to mental health (see Table 1). Only minimal descriptive data on possible activities were provided.

This familiarized participants with the context while still allowing them to think freely, independently and creatively when considering potential solutions. Participants were asked to list any positives or negatives associated with each activity card, as well as any additional considerations or questions they had. Following this, they were asked to rank the activity cards from most to least preferred. After workshop 1—based on the feedback provided from participants and analysis undertaken by the project team—a revised list of eight activity cards (minus “recovery coach” and “on the job gym”) and eight “design principles” (see Table 2) were developed for further consideration in workshop 2. The design principles were the key concepts identified to guide the project design and engagement moving forward and are discussed in detail in Section 3.2.2.

The sensitizing material for workshop 2 included the eight activity cards and eight design principles developed after workshop 1 (following the process of reflection detailed in Section 3.2.3). As in the first workshop, participants were asked to list any positives or negatives associated with each of the activity cards, as well as any additional considerations or questions they had. Differently from workshop 1, participants were asked to consider the eight design principles and rank them in order of perceived importance.

#### 3.2.2. Facilitation

Co-design workshops were held using password-protected meetings on the Zoom platform. All whole-group (comprising all workshop participants) components of the workshop sessions were video and audio recorded with permission. One member of the research team was assigned to record whole group workshop sessions, assign participants to small group “breakout rooms”, monitor session timing and take screenshots of all meeting visuals (e.g., notes, polls, etc.). This researcher was able to communicate directly with individual facilitators, the whole group or specific breakout rooms using the “chat” function in Zoom.


*Workshop 1*


On commencement of the workshop, the participants (*n* = 12) were welcomed to the session and reminded of the aim and expectations for their participation. The group was then split into four virtual breakout rooms which would be used to complete two activities during the workshop. Each breakout room included one facilitator from the research team and three participants.

The first facilitated “breakout” session (30 min) commenced with each participant introducing themselves, their primary production sector and their highest ranked (most preferred) activity. This progressed to a discussion of reasons underpinning ranking decisions. The purpose of this experience was twofold: (i) to develop rapport between group members and (ii) to refresh and extend the sensitization phase. On the completion of this short introductory session, all participants and facilitators returned to the group meeting space. Having discussed and extended their understanding of the activities, participants were invited to share their top three preferred activity cards (from 1–10) and their least preferred activity card using the “live poll” Zoom feature.

The second facilitated session (40 min) tasked the same small groups to design a campaign to reduce risks to mental health for primary producers. Participants were invited to draw on any combination of the ten activities discussed in the first breakout room and/or include completely new ideas. During this time, facilitators only prompted the discussion and did not have any input into the campaign design. On-screen notes were typed in real time by each facilitator, using a pre-designed template and the “shared screen” feature of Zoom. The campaign template included sections to record notes on the activity type, delivery details, promotion method, associated costs, time investment required and campaign name. Upon returning to the main meeting room, a nominated participant from each team delivered a brief “pitch” of their campaign, aided by the completed template, followed by questions from the larger group.

The final minutes of the workshop highlighted the next steps in the co-design process, including an invitation (followed up by an email invitation after the workshop) to a future online feedback session (see Section 3.3.1).


*Workshop 2*


At the commencement of workshop 2, stakeholder participants (*n* = 11) were introduced to the co-design process and provided a summary of the work to date. Participants were also introduced to three members of the digital design team, attending the workshop mainly as observers but available, if required, to provide digital design expertise. This was important to demonstrate the willingness of the researchers and designers to listen closely to stakeholder ideas and concerns around any of the work undertaken thus far in the process. The group was then split into three virtual breakout rooms. Each breakout room included one facilitator from the research team, 3–4 participants and one member of the digital design team. The first session (30 min) explored the design principles posed in the sensitizing survey and provided an opportunity for additional recommendations to be suggested. Having discussed and extended participants’ understanding of the design principles, participants were then invited to share their three most important recommendations using the live poll feature of Zoom.

The second facilitated breakout session repeated the workshop 1 task of designing a campaign to reduce risks to mental health for primary producers. However, in response to the research team’s assessment of workshop 1, it was deemed that the activity timeframe was too short, and an extended time of 60 min was allowed. The groups then returned to the larger group for the pitch presentation and questions.

#### 3.2.3. Reflection

Running multiple online workshops allowed for an iterative co-design loop, enabling tailoring and refinement of the sensitization activities and improvements in workshop facilitation and processes. The data from the first workshop were collated into a detailed draft report by J.M. and then analyzed by other researchers in turn, drawing out recurring themes, as well as similarities and differences across the varying sectors represented. Following discussion by the research team, the data were distilled into eight key design principles (see Table 2) informing project focus, direction and design moving forward. Further data analysis and reflection following workshop 2 led to the addition of a ninth design principle. These nine design principles and the summary of outcomes from workshops 1 and 2 were then made available to the digital design agency team.

#### 3.2.4. Online Considerations for the Workshop Phase

The ability to provide richer pre-workshop sensitization activities that otherwise would often be cramped into the usual single co-design workshop (e.g., [20]) allowed for more primary producer and stakeholder input and ensured sustained levels of engagement without experiencing fatigue. The online environment did not necessarily limit the capacity for interactive methods. For example, participants used virtual “click and drag” tools to rank activity cards and were able to provide rich feedback after exposure to, and consideration of, each of the activity cards. Sensitizing activities familiarized participants with the content and meant that participants—who may have already been anxious about entering the online environment—did not enter the space with the additional stress and expectation of learning new content from scratch. Sensitization ensured familiarity and provided an important vehicle for establishing trust.

Online methods facilitated an increasingly detailed and iterative process of co-design. On-screen note-taking during breakout sessions allowed for clarifications and feedback to happen in real time, with participants being able to correct misinterpretations and expand on the information provided, supporting a richer, more accurate record of the discussion. The iterative process was particularly apparent in the context of running multiple co-design workshops.

### 3.3. Post-Workshop Phase

#### 3.3.1. Reflection and Building for Change

Following the completion of workshop 2, the large pool of quantitative and qualitative data was reviewed and discussed. The team continued to meet at least fortnightly and used deadlines with the design agency to finalize timely reporting and feedback. Reflection remained a significant element of the post-workshop phase, looped in a process of “building for change” (Trischler and colleagues’ step 7) as design concepts were developed and feedback was gathered and processed. The extended report was provided to the digital design team, along with a series of personas (fictional characters) developed by the research team identifying various target audience members. These personas addressed the different needs of users depending on their age, primary production sector, geographic location and sex. The research team worked closely with the design team to identify potential pathways of engagement, resulting in a detailed social network map (produced using digital tools).

A presentation was made by the research and digital design teams to the Advisory Group, highlighting the processes and outcomes of the co-design process and the concept-stage user-driven ideas. This session provided the opportunity for questions and early-stage feedback at a primary producer industry level. The research team continued to work iteratively with the design team, participating in regular online discussion sessions combining visual presentation with feedback and targeted questions. At this stage, a branding team was contracted to develop ideas for marketing and engagement with the target audience of primary producers.

An online feedback session brought together participants from previous co-design workshops as well as other relevant stakeholders (*n* = 37). A summary of work to date was followed by a presentation of the design concept made by the digital design team. This was followed by small group breakout sessions (*n* = 7 groups) facilitated by either a project team or a digital design team member. The discussion sought general feedback on the design concept as well as feedback on targeted areas of interest focusing on user experience and engagement. A presentation of branding concepts (including logo design, name and color palette) was then made by the branding team, with participants being asked to provide general feedback in the chat function of Zoom followed by an online vote for preferred branding via the live poll function of Zoom.

#### 3.3.2. Online Considerations for the Post-Workshop Phase

Given the social distancing restrictions of COVID-19, the research team decided to initially limit the focus on developing engaging strategies to those appropriate for digital delivery. Consideration of opportunities for face-to-face engagement will be delayed until the lifting of restrictions. The presentation of the prototype digital concepts in the feedback session to primary producers and stakeholders was to seek direction and support from users and stakeholders for the final digital solution. This final session also marked an important opportunity to recruit lead users (Community Champions) that are willing to pilot, champion and encourage the uptake of the newly developed intervention.

## 4. Discussion

This paper reported on the process of adapting an existing co-design framework to online delivery as a result of COVID-19. The development of online co-design methods is an important additional research method for anyone (researchers or practitioners) wishing to work with users not only during a pandemic but also when needing to overcome geographic limits or operating with limited resources.

This paper described the application of the seven-step co-design framework (Trischler et al., 2019) in a primary producer context (Victoria, Australia) and reflected on online considerations that researchers should address when co-designing online. Extensions to Trischler and colleagues’ model have also been suggested (see Figure 3), reflective of a multi-workshop, online process.

The extended model emphasizes the following:The iterative process of resourcing, planning and recruiting;The iterative and reflection-driven loop when online methods enable resourcing of multiple co-design workshops;The continuing cycle of reflection and building for change provided through (i) continuing online feedback process, industry engagement and governance and (ii) evolving and flexible design processes.

The results described above have four important implications for future work. First, they demonstrate that co-designing online is possible given adequate preparation, training and resource allocation. Second, they validate that traditionally “hard to reach” populations can be engaged using online co-design methods providing there is adequate relationship building in the pre-workshop phase with the community and relevant stakeholder groups. Third, the results demonstrate that the quality of co-design need not be compromised in the translation of co-design methods from face-to-face to online. Finally, this body of work establishes that resource efficiency and saved costs associated with using online co-design methods can be realigned towards intervention/service creation, promotion and user engagement. We discuss the key advantages and challenges of online co-design next.

### 4.1. Advantages to Translating Co-Design to a Digital Environment

Online co-design during COVID-19 was a process precipitated through necessity, rather than choice. However, numerous advantages have been identified through this transition. Engagement in health and wellbeing services is heavily influenced by a “distance decay” effect in rural areas [21], with increasingly smaller proportions of the population participating as the distance from them increases. This is also likely to affect participation in face-to-face health and wellbeing research. Using online co-design methods has helped to overcome geographic barriers to participation, providing the opportunity to bring together geographically dispersed populations and achieving greater diversity in representation. Similarly, the online co-design process has reduced other barriers to participation (for both researchers and co-design participants), including apprehension about engaging via online methods and barriers associated with the time and cost of travel. Project costs associated with co-design have also been reduced, particularly those associated with researcher travel—allowing for greater investment in other areas of the project.

The swift and widespread onset of COVID-19 necessitated rapid changes in the way services are delivered and the way we communicate. There has been unprecedented adoption of videoconferencing as a way of delivering health and mental health services [22] and as a way of maintaining communication in people’s work and personal lives and in education and training. This shift has forcibly removed previous barriers (technophobia) and increased openness to trying an online engagement approach. This applies to both researchers and participants. For many people, the transition to online interactions has been essential as face-to-face connection has been prohibited due to regulated social distancing, particularly in the state of Victoria—the most impacted state of Australia. This “forced” usage of online engagement has had positive outcomes with researchers and participants developing greater familiarity and comfort in adopting online communication protocols such as muting microphones, camera sharing, chat, virtual backgrounds and using online tools such as whiteboards and live polls. Familiarity with the wide range of available tools allows for engaging and interactive participation and provides increased scope for the use of research methods likely to tap into the range of communication preferences.

Online co-design methods have also helped to break down some of the researcher–participant power imbalances and inequities traditionally noted in previous qualitative research [23]. Having everyone in an online meeting space (all team members and participants) helped to establish participation on a more equal basis. This is particularly notable when comparing to commonly held rural engagement sessions, where some people are centrally located face-to-face and others link in virtually. These “virtual” participants often feel like outsiders in an otherwise face-to-face environment and may also have the added burden of intermittent or poor internet connection [17].

The use of breakout rooms during online co-design sessions presents a range of advantages for both small and large group discussions. For larger groups, breakout rooms can allow for multiple discussions to be held concurrently. Breakout rooms can also be used to engage in more detailed discussions which can then be fed back to the larger group. This method of engagement can also be used to create a more comfortable and intimate environment, providing greater opportunity for participants to share in the conversation and the chance to draw out and engage quieter participants. This can help establish rapport for ongoing engagement in a larger group environment. Online meeting spaces are thought to support engagement in people who struggle with in-person social interaction [24]. This may be particularly important for primary producers, who frequently work alone or in small, family-run businesses and may have limited experience and confidence interacting within large groups. Smaller group discussions in breakout rooms avoid the depersonalization of larger groups and reduce the fatigue elicited by the self-consciousness [25] and performative nature of larger group sessions.

### 4.2. Challenges of Translating Co-Design to a Digital Environment

Adapting co-design to an online environment is not without its challenges. While successful co-design requires researchers to have sufficient resources and expertise to facilitate the engagement process, participants also need sufficient resources and skills to participate fully [26]. In the online environment, this requires capacity and willingness to provide coaching people with minimal online experience in processes from connecting to an online meeting to online etiquette (e.g., muting your microphone to avoid feedback, using the chat function to raise questions rather than speaking over someone). In rural areas particularly, internet connection can be of poor or inconsistent quality. Online co-design planning needs to include responses to these challenges and provide opportunities for participation by phone, or without the use of the camera, if the video connection is poor. Engaging in online co-design is a resource-intensive exercise, requiring a dedicated person to manage the technology and enough facilitators to run breakout rooms. The online process may also produce more data (requiring more analysis), with multiple breakout rooms running simultaneously.

“Zoom fatigue” is becoming an increasingly recognized problem. This is believed to occur for a range of reasons including limited visual cues requiring intense attention [24] and being hyperaware of our on-screen image and how this might be interpreted by others [27]. While Zoom fatigue is anticipated to decrease as people learn to navigate video meetings, online co-design can also reduce fatigue by limiting the amount of time in a single online session and breaking up the activities into different formats (e.g., pre-workshop surveys, different styles of engagement including online polls, virtual brainstorming). Including “transition time”—such as bathroom or drink breaks—between activities allows people to create buffers and refresh during a longer workshop [28]. Finally, actively facilitating and involving participants during small group discussions and providing all participants opportunities to contribute their voice is important to ensure ongoing engagement and avoid disconnection and user fatigue.

## 5. Conclusions 

In summary, while online co-design may have its limitations, it also has many strengths—including the genuine and committed focus of placing the social cohort experience at the forefront—for both the participants and the facilitator [18]. Much can be learned from adapting face-to-face co-design to the online environment. At this stage, it seems unlikely that online co-design will replace all face-to-face co-design in the future. However, elements of this newer online co-design can no doubt inform more effective processes in the face-to-face environment once COVID-19 social distancing restrictions are lifted. This should have particular benefits for advancing the co-design of interventions/services for both rural and city-based communities in high-income countries and for projects where resources are limited but co-designed solutions are of interest. This lays a firm foundation for greater recognition and value of inclusion on an equal participatory basis.

## Figures and Tables

**Figure 1 ijerph-18-04147-f001:**
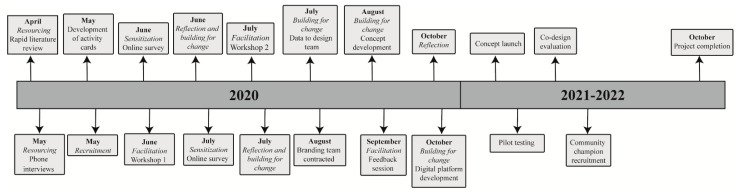
Project timeline.

**Figure 2 ijerph-18-04147-f002:**
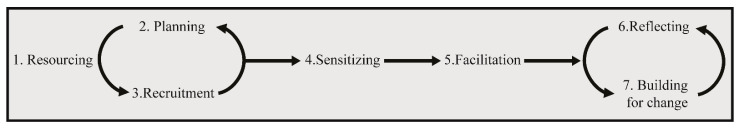
Trischler and colleagues’ seven-step co-design framework. Figure adapted from Trischler et al., 2019.

**Figure 3 ijerph-18-04147-f003:**
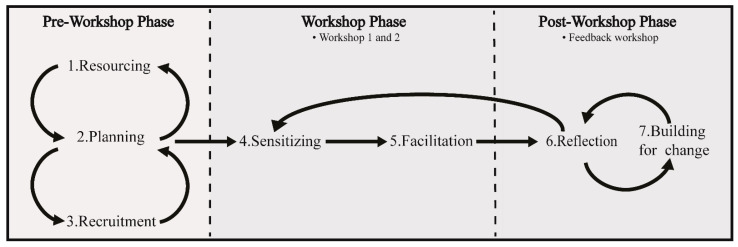
Extended model of Trischler and colleagues’ co-design framework.

**Table 1 ijerph-18-04147-t001:** Developed activity cards.

Activity Card	Description
**Primary Producer Success Stories**	Aims to transfer knowledge and inspire positive action
**Adapting to change**	Aims to develop personal and business-related resilience skills to help overcome personal and business-related challenges
**The resource hub**	Aims to provide information and links to new and existing support resources and services to provide a one-stop-shop destination
**On the job gym**	Aims to encourage physical activity (and social connection where possible) to improve/build physical and mental strength
**Peer mentoring network**	Aims to transfer knowledge and experience and encourage personal networks
**Your business toolkit**	Aims to provide practical resources to assist with proactive business planning to ensure business success
**Establishing your local network**	Aims to support primary producers in establishing/building a local community network for the purpose of capacity building and social connection
**Hire an expert**	Aims to provide access to one-on-one expert support for primary producers
**Recovery coach**	Aims to help improve sleep and rest efficiency to increase mental, physical and business health

**Table 2 ijerph-18-04147-t002:** Design principles developed after workshop 1 and revised after workshop 2.

1.	**Personal connection is essential:** Personal stories, local linkages and social connection are all important.
2.	**Keeping an eye on the goal (prevention of risks to mental health):** Strategies must maintain focus on prevention of risks to mental health.
3.	**Language matters:** Language used needs to (i) avoid stigmatization and stereotypes and (ii) reflect understandings of primary producers as goal-focused and pragmatic.
4.	**One size will not fit all:** Strategies need to reflect the varied primary production sectors, digital connectivity, expertise, needs, age groups and geographic locations of target participants.
5.	**There is limited downtime as a primary producer:** Primary production schedules are busy, with peak seasons of intense workload. Strategies should be achievable in manageable steps and incorporated into work routines and practices.
6.	**Local matters:** Local experience and knowledge are often privileged. Focusing on local people, knowledge, experience and expertise will support meaningful engagement that is well timed, relatable and useful.
7.	**Personalization ensures engagement:** Primary producers need strategies that are convenient and tailored to their needs. While unique strategies for individuals may not be possible, individuals need to feel understood and be able to identify a pathway that is right for them.
8.	**Virtual/digital connection is becoming more acceptable and accessible in a range of formats:** Opportunities for digital engagement are becoming increasingly accepted and varied. Strategies must cross a range of mediums.
9.	**Utilize knowledge, resources and networks that are already in existence ^1^:** Various local networks and resources for primary producers already exist. PPKN needs to tap into existing networks first and extend from there to those that are harder to reach. PPKN needs to act as a conduit to existing materials and resources.

^1^ Added following reflection after workshop 2.

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
