# Peer review of "Translating Co-Design from Face-to-Face to Online: An Australian Primary Producer Project Conducted during COVID-19"

_ijerph, 2021, doi:10.3390/ijerph18084147_

Round 1

Reviewer 1 Report

In the introduction or theoretical framework that supports this article, I consider that it could be improved by incorporating the following references:
Flores Cuevas, F. (2018). Pedagogical training and the use of information and communication technologies within the teaching-learning process as a proposal to improve their teaching activity. EDMETIC, 7 (1), 151-173. https://doi.org/10.21071/edmetic.v7i1.10025
Escudero Nahón, A., & Núñez Urbina, A. A. (2019). Theoretical foundations for the transformation of the "Massive Open Online Courses" towards "Customizable Open Online Courses". EDMETIC, 8 (2), 129-149. https://doi.org/10.21071/edmetic.v8i2.10988
Marín-Díaz, V., Riquelme, I., & Cabero-Almenara, J. (2020). Uses of ICT tools from the perspective of Chilean University Teachers. Sustainability, 2 (15), 6134, doi: https://doi.org/10.3390/su12156134
In the methodology section, it should be indicated why the authors of the article themselves, who are also the project researchers, were selected as a sample.
The participant selection process is not clear to me, I suggest justifying it with a bibliographic reference.
Some section on elearning should be included, since the situation at the Australian level is not clear

Author Response

REVIEWER 1:

Suggested Change

Author Comments

1. In the introduction or theoretical framework that supports this article, I consider that it could be improved by incorporating the following references:
Flores Cuevas, F. (2018). Pedagogical training and the use of information and communication technologies within the teaching-learning process as a proposal to improve their teaching activity. EDMETIC, 7 (1), 151-173. https://doi.org/10.21071/edmetic.v7i1.10025
Escudero Nahón, A., & Núñez Urbina, A. A. (2019). Theoretical foundations for the transformation of the "Massive Open Online Courses" towards "Customizable Open Online Courses". EDMETIC, 8 (2), 129-149. https://doi.org/10.21071/edmetic.v8i2.10988
Marín-Díaz, V., Riquelme, I., & Cabero-Almenara, J. (2020). Uses of ICT tools from the perspective of Chilean University Teachers. Sustainability, 2 (15), 6134, doi: https://doi.org/10.3390/su12156134

1. Thank you for the suggested references. Unfortunately, the full text of the first two references are available only in Spanish and cannot be read by any members of our team. The third reference has been referenced and we thank the reviewer for their suggestion (see page 7). We have also included some other comments around online education and training.

Marín-Díaz, V., Riquelme, I., & Cabero-Almenara, J. (2020). Uses of ICT tools from the perspective of Chilean University Teachers. Sustainability, 2 (15), 6134, doi: https://doi.org/10.3390/su12156134

2. In the methodology section, it should be indicated why the authors of the article themselves, who are also the project researchers, were selected as a sample

2. In qualitative research, the researcher is considered integral to the research process. The following has been added to clarify this:

Acknowledging the reflexive relationship between researchers and the research process, consideration of the researchers’ role is considered imperative (Jootun, McGhee, & Marland, 2009). 

3. The participant selection process is not clear to me, I suggest justifying it with a bibliographic reference.

3. The following bibliographic reference has been added to increase understanding of the purposive sampling strategy used to select participants:

Palinkas LA, Horwitz SM, Green CA, Wisdom JP, Duan N, Hoagwood K. Purposeful Sampling for Qualitative Data Collection and Analysis in Mixed Method Implementation Research. Adm Policy Ment Health. 2015;42(5):533-544. doi:10.1007/s10488-013-0528-y

4. Some section on elearning should be included, since the situation at the Australian level is not clear

Whilst this paper is around adapting co-design to an online process and format there are similarities with well-designed online learning.  These have been included on page 7 and 8 along with a short comment on the situation with the online learning level during COVID-19 in Victoria, Australia particularly.

Reviewer 2 Report

This article is an interesting and well written description of how the product Zoom is used to conduct user involving innovation It may inspire others to do the same and provide information which others can learn from.

As a research article, it is difficult to find the innovation or even research in the work. All suggestions are today familiar to educators and others using virtual tools for interacting with students and colleagues and the suggestions and activities can be found in the material from higher education learning labs. The reviewer lacks e.g., reports on trial and comparison of various ways to work and how various software products can be used. In particular it is important to note that Zoom is a very strong tool for direct interaction but in 2020 was not the state of the art for collaboration. Some features have now been added, but in 2020 it needed to be combined with other platforms and also at that time Microsoft Teams had more features for documentation etc. The reviewer therefore needs a more thorough description of why Zoom was chosen and especially how the security issues were addressed. In relation to Zoom it also needs a little more reflection and description and even demonstration of various models in relation to how the supporters acted and how “break out rooms” was used. Why were people allocated and not asked to freely mix or shop around in the rooms? There is a lot which could be explored and the tests described.

Did the investigators have any research questions with respect to how the virtual interaction would influence people’s behavior and did they actually explore this and how did they assess the participants experiences and accept and did they try to relate the adherence and user experience to e.g. theories of technology acceptance, ehealth literacy or digital literacy?

The reviewer find that the current report probably cannot be improved much based on the way it is written, it is just about whether the scope and intention of the text will suit the Journal. The reviewer hope with these comments to inspire to a research study which could either address the documentation of effects of new ways to work and explore new paths or go deeper into the user experience perspective.

Author Response

REVIEWER 2:

Suggested Change

Author Comments

This article is an interesting and well written description of how the product Zoom is used to conduct user involving innovation It may inspire others to do the same and provide information which others can learn from.

1. As a research article, it is difficult to find the innovation or even research in the work. All suggestions are today familiar to educators and others using virtual tools for interacting with students and colleagues and the suggestions and activities can be found in the material from higher education learning labs. The reviewer lacks e.g., reports on trial and comparison of various ways to work and how various software products can be used. In particular it is important to note that Zoom is a very strong tool for direct interaction but in 2020 was not the state of the art for collaboration. Some features have now been added, but in 2020 it needed to be combined with other platforms and also at that time Microsoft Teams had more features for documentation etc. The reviewer therefore needs a more thorough description of why Zoom was chosen and especially how the security issues were addressed. In relation to Zoom it also needs a little more reflection and description and even demonstration of various models in relation to how the supporters acted and how “break out rooms” was used. Why were people allocated and not asked to freely mix or shop around in the rooms? There is a lot which could be explored and the tests described.

1. Zoom was the chosen online platform as the lead research institution held a licensed agreement and widely implemented its use during COVID-19. The following has been added to clarify this (page 5):

Zoom was the chosen communications platform due to its demonstrated reliability in providing quality video and online conferencing functionality. The lead research institution also held a licensed account, enabling access to security functions such as password protected meeting spaces

For the requirements of the co-design workshops and the planned activities, Zoom offered all of the features required by the research team, including, breakout rooms, poll features, recording capabilities and the ability to share screens. There was therefore no need to investigate additional platforms. A statement to this effect has also been included on p5

Details of how the breakout rooms were used has previously been described throughout the results section of the manuscript, as follows:
The first facilitated ‘breakout’ session (30 minutes) commenced with each participant introducing themselves, their primary production sector and their highest ranked (most preferred) activity. This progressed to a discussion of reasons underpinning ranking decisions. The purpose of this experience was twofold: (i) to develop rapport between group members, and (ii) refresh and extend the sensitization phase. (Page 9)

The second facilitated session (40 minutes) … On-screen notes were typed in real time by each facilitator, using a pre-designed template and the ‘shared screen’ feature of Zoom. The campaign template included sections to record notes on the activity type, delivery details, promotion method, associated costs, time investment required, and campaign name. (page 9)

The first session (30 minutes) explored the ‘design principles’ posed in the sensitizing survey and provided opportunity for additional recommendations to be suggested.

The second facilitated ‘breakout’ session repeated the workshop one task of designing a campaign to reduce risks to mental health for primary producers.

Participants were allocated into the virtual breakout rooms as it allowed rapport to be developed between group members (page 9) which created a more comfortable and intimate environment, providing greater opportunity for participants to share in the conversation and the chance to draw out and engage quieter participants (Page 14). If participants were given the ability to freely mix or shop around between rooms, the limited time available to complete the activities would have been spent completing introductions and data collection would be limited.

2. Did the investigators have any research questions with respect to how the virtual interaction would influence people’s behavior and did they actually explore this and how did they assess the participants experiences and accept and did they try to relate the adherence and user experience to e.g. theories of technology acceptance, ehealth literacy or digital literacy?

2. The purpose of the codesign workshops was to identify strategies that would be of interest and meet the needs of primary producers on issues affecting their mental health in the workplace. The workshops were not aimed at influencing the behavior of the participants.  However, virtual interaction and behavior is relevant in the next stage of the study—building for change—which involves the development and delivery of a digital platform. This platform will be evaluated using both quantitative (surveys) and qualitative (focus groups) methods to explore users’ engagement with the platform including acceptance, usefulness and ease of use. Analysis of this data will include theories of technology acceptance and other digital health related literature.

3. The reviewer find that the current report probably cannot be improved much based on the way it is written, it is just about whether the scope and intention of the text will suit the Journal. The reviewer hope with these comments to inspire to a research study which could either address the documentation of effects of new ways to work and explore new paths or go deeper into the user experience perspective

3. N/A

Reviewer 3 Report

Overall, a topical, interesting paper that highlights research challenges both in a rural context and also in a pandemic setting. Some minor comments below. 

  1. The introduction was good, utilising relevant literature that supported the project. It would be beneficial to have a little more emphasis on research design rather than the mental health aspect for primary producers.
  2. The methods section was clear, however contained very little information about how the study was carried out. The results/methods sections are quite blurred. I recommend clarifying what the "results" are. I was expecting to read abut how co-design led to improved initiatives and themes around preventing risks for mental health in rural Victoria. And all of the results section was about the method. e.g. recruitment, phases etc... It would be beneficial to expand the methods section and refine the results. 
  3. The discussion was sound and provides some great recommendations for future work in an online space for co-design research. The extended model was also a good way to demonstrate this. 
  4. Your conclusions aligned nicely with the limitations for rural areas participating in research and was also realistic in its discussion of face to face vs online. 

Author Response

REVIEWER 3:

Suggested Change

Author Comments

Overall, a topical, interesting paper that highlights research challenges both in a rural context and also in a pandemic setting. Some minor comments below.

1.The introduction was good, utilising relevant literature that supported the project. It would be beneficial to have a little more emphasis on research design rather than the mental health aspect for primary producers.

1. Thank you for your positive comments on our introduction and literature review. In line with your suggestion, we have removed some background on the mental health aspect for primary producers in favour of more emphasis on the research design and the rationale for this paper.  The second paragraph of the introduction now reads:

The COVID 19 pandemic has hastened citizens’  use of the Internet with usage increasing from between 40% to 100%, compared to pre-lockdown levels (De', Pandey, & Pal, 2020). Researchers have needed to pivot online to continue their citizen-focused research due to the limitations caused by the pandemic. All co-creation methods, including co-design, have historically strongly relied on face-to-face engagement and translating these methods into an online format is now urgently warranted. This point in time presents a valuable opportunity for exploring new ways of empowering primary producers to work with researchers to identify the challenges and co-design the best solutions (D'Proff, Mills, & Gray, 2020). This also provides the opportunity to fill an important gap for co-design research and demonstrate how co-design can be executed fully online using a sequential and evolving series of co-design workshops following the Trischler et al. process. This paper makes important theoretical and methodological contributions, guiding future researchers and practitioners to run sequential co-design sessions online with multiple and cross-sectoral stakeholder groups.

2. The methods section was clear, however contained very little information about how the study was carried out. The results/methods sections are quite blurred. I recommend clarifying what the "results" are. I was expecting to read about how co-design led to improved initiatives and themes around preventing risks for mental health in rural Victoria. And all of the results section was about the method. e.g. recruitment, phases etc... It would be beneficial to expand the methods section and refine the results

2. We have now edited our abstract, the research questions in the introduction and the first paragraph of the results, to make it clearer that the adaptions made to Trischler and colleagues’ co-design method (to make it work in an online environment with primary producers), are in fact an important part of our results. Thank you for bringing this issue to our attention. We believe these changes have enhanced the clarity of the paper.

3. The discussion was sound and provides some great recommendations for future work in an online space for co-design research. The extended model was also a good way to demonstrate this

3. Thank you

4. Your conclusions aligned nicely with the limitations for rural areas participating in research and was also realistic in its discussion of face to face vs online

4. Thank you

Round 2

Reviewer 1 Report

I believe that the modifications made by the authors have enriched the article